# Social Skills Group Training for Students with Neurodevelopmental Disabilities in Senior High School—A Qualitative Multi-Perspective Study of Social Validity

**DOI:** 10.3390/ijerph19031487

**Published:** 2022-01-28

**Authors:** Emma Leifler, Christina Coco, Anna Fridell, Anna Borg, Sven Bölte

**Affiliations:** 1Center of Neurodevelopmental Disorders (KIND), Centre for Psychiatry Research, Department of Women’s and Children’s Health, Karolinska Institutet, 113 30 Stockholm, Sweden; christina.coco@ki.se (C.C.); anna.fridell@ki.se (A.F.); anna.borg@ki.se (A.B.); 2Stockholm Health Care Services, Region Stockholm, 171 77 Stockholm, Sweden; 3Department of Pedagogical, Curricular and Professional Studies, University of Gothenburg, 405 30 Gothenburg, Sweden; 4Child and Adolescent Psychiatry, Stockholm Health Services, Region Stockholm, 171 77 Stockholm, Sweden; 5Curtin Autism Research Group, Curtin School of Allied Health, Curtin University, Perth, WA 6102, Australia

**Keywords:** social skills group training, autism, ADHD, neurodevelopmental disabilities, inclusive education, school, social validity

## Abstract

Including students with neurodevelopmental disabilities (NDDs) in regular classrooms has become a law-enforced common practice in many high- and middle-income countries. Still, without appropriate actions supporting the implementation of inclusive pedagogical practice, students with NDDs remain at increased risk for absenteeism, bullying and underachievement. There is limited knowledge on the feasibility of social skills group training (SSGT) in naturalistic settings. Using a qualitative approach, the objective of this study was to explore the lived experiences of (i) students diagnosed with autism or attention-deficit hyperactivity disorder and those showing subclinical social difficulties receiving either SSGT or active social control activities in a regular senior high school setting, (ii) teachers providing SSGT or the active control activity and (iii) school leaders facilitating the implementation of these actions. Due to the impact of the COVID-19 pandemic, comparison between real life versus digital administration of SSGT was also examined. Within a randomized controlled pilot trial of the school-tailored SSGT SKOLKONTAKT^®^, the primary perspectives of 20 students, teachers and school leaders on SSGT or the social control activities were explored. All groups perceived SSGT to enhance school attendance and academic achievement of students, as well as teacher inclusion skills and the social school climate. Findings indicate that SSGT is largely feasible and socially valid, and broader implementation of SSGT in school settings appears meaningful.

## 1. Introduction

Neurodevelopmental disabilities (NDDs) are common conditions arising from variations of the function, structure or maturation of the developing brain leading to cognitive alterations and functional impairments in important areas of daily life [1,2,3,4,5]. NDDs are associated with reduced quality of life, increased risk of psychiatric and somatic complications and premature mortality [6,7]. In educational settings, students with NDDs, such as autism spectrum disorder (ASD) and attention-deficit hyperactivity disorder (ADHD), are significantly overrepresented among those affected by bullying, school-related anxiety, loneliness and absenteeism as well as poor academic attainment [8,9,10,11].

Social communication and social interaction expectations and demands are immanent to school life. However, most NDDs, especially ASD, and, to a lesser degree, ADHD, are defined by or associated with social challenges (e.g., initiating and engaging in conversations and social regulations), in different areas of life, including school life [12,13,14,15]. This limits the opportunities of individuals with these conditions to learn and improve social skills and increases risk of social isolation and withdrawal [16]. Generally, social skills are associated with the quantity and quality of friendships [17], which in turn are associated with happiness and quality of life [18]. School is a social arena where peer likeability is positively associated with motivation, satisfaction and interest in school and scholastic performance [19,20]. Particularly in adolescence, engaging in social interaction with peers is a significant element of daily life in school [21].

NDDs are increasingly viewed as non-pathological, dimensional neurodivergent human phenotypes [22,23], and the understanding and support of individuals diagnosed with NDDs is not only the duty of clinical and special education staff, but of society as a whole, including mainstream educational settings. Therefore, particularly high- and middle-income countries have introduced inclusive education as a mandatory element of their regular school systems [24,25,26,27]. If provided with appropriate support in the learning environment, students with disabilities benefit from typical educational settings [28,29]. The inclusive setting is an accommodated learning environment where students with disabilities benefit from learning alongside their neurotypical peers. Consistently, the recently published Lancet Commission on future care and clinical research in autism concluded that naturalistic environments are the most appropriate setting for interventions [30]. In addition, it demonstrated that research has disproportionately focused on early development, while research on adolescence is only rarely studied. Thus, intervention research in school settings in adolescent populations provides a great opportunity to both bridge existing research gaps and increase the building of capacity for people with NDDs in important arenas of society, beyond clinical settings.

In the narrow and ambitious definition, inclusive education seeks to achieve accommodations of the regular school environment to the diverse prerequisites and needs of all students, not demanding the student to adapt to a given environment [31]. A broader definition of inclusion also welcomes actions directed to the student, if their primary purpose serves improved skills which foster learning opportunities and empowerment in the regular classroom for individuals or groups who have traditionally been excluded or are typically disadvantaged or stigmatized. For instance, this definition of inclusive education is used by UNICEF [32] when describing that all children with disabilities have the right to quality education and learning. The broader definition, therefore, values targeted interventions for students that support their development and acquisition of skills to interpret and cope with activities and demands occurring within the school day. Looking at inclusive education with holistic lenses include analyzing the environment and reducing obstacles in it in order to meet diverse prerequisites. One needs to take into account all dimensions of the learning environment, as well as how individuals interact and are affected by environments.

Currently, most mainstream schools across countries still struggle to make inclusive educational practice a reality for students with NDDs [33,34]. This is unfortunate, as students with NDDs who are not supported adequately in mainstream settings either benefit less from being placed among typically developing peers or are at risk for even greater social exclusion [35,36]. Studies on actions consistent with a narrow definition of inclusion, that is accommodations in the learning environment to the needs of students with NDDs, are scarce and have not yielded conclusive results specific to students with NDDs [37,38,39]. On the other hand, systematic reviews and meta-analyses on school-based interventions directed to students with NDDs have shown positive effects on academic and behavioral outcomes, e.g., for students with ASD and ADHD [40,41].

Social skills group training (SSGT) is an umbrella term for interventions applying socially instructive techniques and behavioral modification principles in group settings to improve social skills. In clinical settings, SSGT is widely used in pre-school and school-aged children with ASD and have demonstrated robust and moderate effects in participants with average or above-average general cognitive abilities (IQ > 70) in two systematic reviews and meta-analyses [42,43], and subsequent large randomized controlled trials [44,45]. Increased social skills have been reported, as well as a decrease in autistic-like traits and social anxiety. In ADHD, SSGT is less developed and established compared to ASD and more often embedded into intervention programs of wider scope, also targeting other skills, such as meta-cognition, daily life and organizational skills [46,47]. A systematic review of social training in ADHD [47], mostly reporting on SSGT, found positive effects on various aspects of social functioning, although methodological issues and the less social skills-focused nature of the interventions make it difficult to judge the value of SSGT as a stand-alone intervention in ADHD. Reviews as well as qualitative and quantitative studies on clinically delivered SSGT for youth with ASD and ADHD stress research and practice should consider SSGT to be implemented by classroom teachers in naturalistic school settings. Benefits may include practicability for participants and families, facilitation of generalization, lower thresholds for participation, prevention of negative mental health outcomes, better ecological validity or possible spin-off in terms of qualification of staff and better social relations between staff and students [41,44,48,49,50,51]. Implementation of SSGT in mainstream school settings would also comply well with inclusive education legislations for students with NDDs [52,53].

Although the literature on social skills training in various school settings and groups of students is promising regarding the effects on social and emotional skills, school attitudes and behavior and academic performance [54,55,56,57,58,59], especially in more recent meta-analyses, there are still issues concerning scope and application (e.g., creating effective interdisciplinary collaborative teams, allocation of the responsibility for implementation and finding time in busy schedules). The same is true for SSGT directed to students with NDDs. In a recent meta-analysis on social skills training for autistic students in inclusive school settings, Dean and Chang [60] found 18 studies indicating collectively that school-based social skills interventions are effective in improving social outcomes for autistic students in childhood. However, social training approaches for NDDs in schools predominantly used individual training, social clubs, social collaboration, peer-mentoring, social stories, video modeling and combined interventions using specific social skills training elements, predominantly provided by health care professionals or researchers in children [10,47,60,61]. More explicit and comprehensive SSGT administered by school staff in adolescents with NDDs has not been studied. Moreover, while data from children indicate that social training might not necessarily be more effective in a group setting than in individual administration in NDDs [62], group formats are often more cost-effective in youth [63,64], can serve far more students at the same time and are probably more consistent with the group teaching culture at schools. Moreover, given the solid evidence for explicit SSGT in NDDs from clinical studies, research indicating that adolescents with NDDs may require age-appropriate training and intervention goals other than children [65] and that effects of social training are perhaps higher in school when provided by school staff [10], studies on SSGT for adolescent students with NDDs in regular school settings conducted by teachers are desirable. From a societal capacity building point of view, the latter also appears significant to achieve long-term sustainability and implementation of social skills training in naturalistic school environments.

An obstacle to the implementation of teacher facilitated SSGT for adolescent students with NDDs in mainstream schools might be limited feasibility or lack of compliance by students, teachers or school leaders. It could be deemed practically challenging to organize SSGT in the organization, owing to time and space restrictions, or its synchronization with other prioritized activities and duties. Students are perhaps not motivated to participate, and their can exist fears to experience aggravated stigma if interventions are conducted in their natural daily environment and in proximity to neurotypical peers. In addition, students could refuse teachers as SSGT facilitators, due to previous conflicts or mistrust. Teachers and school leaders may be pessimistic about their own capacities to conduct or being responsible for “clinical-like” activities or feel that SSGT does not lie within the mission of an educational institution. The principles of SSGT might not be sufficiently consistent with the school staff’s professional traditions and philosophies so that reaching allegiance to the method could be an issue [66]. In face of these potential barriers to SSGT implementation in regular school, investigating and ensuring social validity of the intervention in the target educational environment appears essential. Social validity refers to an assessment of consumer acceptability and satisfaction of an intervention, a significant prerequisite for the long-term success of any action. Social validity can be understood as a pivotal aspect of external validity, the degree to which an intervention works in reality as opposed to typically more controlled research environments. Unfortunately, external validity has been poorly reported in the SSGT literature [67], and participants attitudes and experiences have been rarely reported [50,68,69].

In conclusion, the implementation of SSGT for students with NDDs in regular school appears both scientifically, societally and economically promising and is consistent with educational policies about inclusive education. However, studies conducting stand-alone SSGT in adolescents with NDDs and teachers providing the training in mainstream school settings are scarce. In addition, little is known about the social validity of SSGT under such prerequisites. Finally, it is unknown if SSGT is perceived as different from other socially engaging group activities in school context. Therefore, the objective of this study was to examine the social validity of SSGT compared to other social group activities in a mainstream context administered by school staff to adolescents with NDDs. Due to the COVID-19 pandemic emerging during the study, we also examined how the target groups experienced real-life and digital formats of delivered training. A qualitative design was applied capturing the perspectives of adolescent participants, teachers and other staff providing the training and school leaders facilitating the implementation of SSGT at their respective schools.

## 2. Materials and Methods

### 2.1. Participants and Settings

The study was approved by the national ethics authority and oral and written informed consent was acquired prior to study participation. The sample for this qualitative multi-perspective study of social validity was recruited purposefully with the goal to identify and include as many information-rich cases who had been actively involved in the implementation and evaluation of the SSGT SKOLKONTAKT^®^ in a mainstream senior high school in Stockholm, Sweden. Prior to the SSGT implementation and evaluation study, teachers and school management had identified students who could benefit from the intervention and had approached and informed the caregivers and their adolescent students about the opportunity. The informed consent also included a request about providing or allowing access to copies of their clinical records.

This study comprised a total of *n* = 20 participants divided into three groups with specific perspectives: 13 adolescent students with NDDs who had received either SSGT (*n* = 6) or an active social control activity (*n* = 7) (Table 1), three teachers and two social workers from the school (in the following called teachers), who had provided SSGT to the students, and two principals (school leaders), who had facilitated the evaluation and implementation of SSGT at the respective school. Adolescent students with NDDs or related subclinical social challenges (self-identified as five female, five male and three diverse) were aged between 17 and 20 years (*Md* = 18 years) and had average to above average intellectual abilities (all IQ > 70). All students except for one had community primary diagnosis of ASD or ADHD, and one had subclinical autistic symptoms (with a primary diagnosis of social anxiety). Students had been diagnosed within child and adolescent psychiatry services according to ICD-10 criteria and in line with the clinical diagnostic guidelines of Region Stockholm [70]. Teachers providing SSGT all had on average 15 years of education experience. The teachers were familiar with the participating students prior to enrollment in the training.

### 2.2. Design and Procedure

In order to establish the social validity of SSGT in a school setting, this study used a qualitative in-depth interview design to collect the lived experiences and attitudes of multiple groups of individuals (adolescents with NDDs, teachers and school leaders) participating in SSGT or its planning, implementation or administration. The qualitative study was conducted during and after a registered quantitative randomized controlled pilot trial of SSGT KONTAKT^®^ (clinicaltrials.gov, identifier NCT04302818). In the trial, 33 adolescent students with NDDs or subclinical social challenges were randomized to either SSGT (*n* = 17) or a social activity control condition (*n* = 16). The groups were conducted by four high school teachers and two social workers employed at the school, headed by two school leaders. Thus, 48% of all adolescent participants, 83% of all school staff and all school leaders involved in the quantitative pilot study also took part in the current qualitative study on the SSGT program’s social validity. The primary outcome in the pilot randomized controlled trial are changes in self-, teacher- and parent-reported social skills according to the Social Skills Group Questionnaire. Secondary outcomes include quality of life, achievement of personally meaningful social goals, autistic trait severity, adaptive functioning and negative side effects. Measures were taken at baseline, 12 weeks post SSGT and at a 3-month follow-up. In-depths interviews were conducted with the participants post SSGT assessment. The pilot study was conducted in three waves over three semesters between August 2019 and January 2021. SSGT was delivered face-to-face at school in wave 1. In wave 2, due to the effects and restrictions of the COVID-19 pandemic, the SSGT training and control activities were transformed to digital format half-way, and conducted as an online training with school staff being placed at personal computers at school and adolescent students participating via personal computers from home via Google Meet. In wave 3, SSGT digital and real-life formats were mixed.

### 2.3. Interventions

#### 2.3.1. Social Skills Group Training

Students in the SSGT group received SKOLKONTAKT^®^ (Swedish for school-contact) [71], a structured and manualized SSGT training for children and adolescents with social communication and social interaction challenges designed for school settings, developed in Sweden. The target group consisted of students with subclinical and clinical social challenges associated with social functioning impairments, often on the ASD and ADHD spectrum. SKOLKONTAKT^®^ has been derived from KONTAKT^®^ [72], an SSGT for clinical settings that has demonstrated the feasibility and effectiveness in previous research [67,73,74,75], and has been adapted to the specific prerequisites and requirements of the school environment. In SKOLKONTAKT^®^, compared to KONTAKT^®^, participants do not require a clinical diagnosis, and they receive training in their natural daily educational environment. In addition, SSGT is conducted by regular school staff, not clinicians, although school staff is trained and supervised by clinicians (usually certified psychologists experienced in NDDs and cognitive behavioral therapy) for quality control. The training is delivered in shorter sessions of higher frequency and the content is more school-focused. Through the adaptation of the training to school context, SKOLKONTAKT^®^ seeks to build capacity and competence among teachers and schools, increase the proximity of training to real life challenges, enhance generalization and increase accessibility by lowering the threshold for participation practically. The character of SKOLKONTAKT^®^ is, thus, more preventive than its clinical counterpart.

In addition to the ongoing pilot RCT of SKOLKONTAKT^®^ to which this qualitative study was attached, a recent small-scale pilot study demonstrated promising feasibility in terms of attendance and satisfaction with SKOLKONTAKT^®^ among students and teachers in a small open pilot-study (*n* = 7) at a high school in the municipality of Strängnäs, Sweden (unpublished data).

The characteristics and formats of SKOLKONTAKT^®^ are shown in Figure 1 and Table 2. Briefly, the usual SSGT size is four to eight students and two to three school staff members. The training is mostly based on cognitive behavior therapeutic principles, such as behavior activation, psychoeducation, cognitive training and observational learning. The training principles are embedded into the mandatory, recurring and variable elements of SKOLKONTAKT^®^ that are guided and conveyed by the trainers. All elements seek to challenge and restructure participants’ thoughts and opinions about social constructs, communication processes and the participants’ skills to manage them, particularly regulating their emotions and actions. The duration of the study was 12 weeks, with three sessions per week lasting 50 min (36 sessions in total). The digital variant during COVID-19 lockdown included three sessions of 40 min plus separate short individual coaching sessions and weekly homework. In SKOLKONTAKT^®^, school staff receive an introductory training followed by monthly supervision. Staff was provided with extra supervision during transformation to the digital version. SKOLKONTAKT^®^ sessions follow a structured format, starting with an opening round and finishing with a closing round. Recurring activities included theme-based group discussions, psychoeducative elements and weekly social missions as well as individually formulated social goals. In addition, participants conduct social group practice activities, including role play and emotion recognition training. Examples of participants’ specific social communication goals at school may include answering questions by the teacher in the classroom or starting conversations.

#### 2.3.2. Social Activity Control Group

Students in the active control group took part in social group activities, such as playing board games, baking or sportive activities, not including explicit social training elements. The control group followed the same format regarding group size, staff ratio, frequency, location and lengths sessions. The social activity control group did not follow a strictly standardized program and was led by teachers.

### 2.4. Interviews

To collect verbal data on the social validity of SSGT and active social control activities in regular school settings, we used structured interview guides specifically designed to tap the lived experiences of students, teachers and school leaders having a role in receiving, administering or facilitating SSGT or the active social control activities. Interview guides were developed for the different groups of informants (adolescent students, teachers and school leaders) based on the work of Wolf [76] on social validity, Choque Olsson et al. [50] on the lived experience of individuals participating in KONTAKT^®^ and Jonsson, Choque Olsson and Bölte [67] on the external validity of social skills training. The interview guides contained open questions that were tailored to the roles of the participants, and all were meant to stimulate the responders to reflect on positive and negative indicators of SSGT in their natural school environment. Adolescents were asked to reflect upon how they remembered and perceived SSGT, which parts they liked and disliked, which could be improved or changed, which they found helpful, or whether SSGT had changed their social skills and behaviors. They were also inquired about the duration of the training and implementation in their school lives. Teachers were asked similar questions on the contents, structure and duration of SSGT/control activity for examining social validity, if it had improved the student’s social interaction skills and the quality of their own interaction with them, as well as their own experience of competence. Moreover, they were asked if they found the implementation of SSGT/control activities in school generally useful and realistic to be included in the school’s educational concept. School leaders were interviewed about the process of implementing the training and taking part in the research project, the need of such a training in regular school of the SSGTs appropriateness for this purpose, how the SSGT had impacted their school, teachers and students, if it was a realistic tool for their school and what about the SSGT and its implementation could be improved. See Table 3 for a detailed summary of the interview guides. The questions were identical for the control and active training group, whenever possible (e.g., differed only for questions on activities specific to each group).

The interviews were conducted by the first author (E.L.). The interviewer has long-term previous experience in special education working with adolescents diagnosed with NDDs or showing other needs, as well as educating and supervising teachers in these issues. To avoid investigator bias, the interviewer was completely independent from the quantitative study, training of students, training and supervision of teachers and development of SKOLKONTAKT^®^ and its implementation at the respective school. The average duration of the interviews with the students, school staff and school leaders was 40 min. The duration of the interviews with students was markedly shorter (~15 min) for those students who had received digital SSGT or social control activities. In the first wave, interviews took place in a quiet room at the school. The study had an ethnographic approach, with the interviewer having spent time in the school prior to the interviews being conducted aiming to build trust and a more secure interview experience for the participants. During the second and third wave, due to COVID-19 restrictions, the interviews were conducted on Google Meets, with the adolescent present at school or at home and the researcher interviewing them remotely.

### 2.5. Qualitative Data Analysis

Interviews were audio recorded, transcribed verbatim and then coded to condense the material into consistent emerging themes using thematic analysis [77,78] in a standard multistep fashion: (i) generating initial codes, (ii) collating codes into potential themes, gathering all data relevant to each theme, (iii) defining and naming themes, generating a thematic map of the analysis, (iv) generating clear definitions of each theme, (v) final analysis and interpretative process with the research questions and literature and (vi) comparison between themes in SSGT versus social activity control group. The process of translating the transcribed data into small units of codes and translating codes into building blocks for themes and subthemes representing patterns of meaningful core ideas was facilitated by using NVivo 12 (QSR Ltd., Burlington, VT, USA). The material was further interpreted in accordance with the socioecological framework by Bronfenbrenner [79], where the student is viewed interacting with its environment. Bronfenbrenner’s model holds a key proposition: in order to develop skills, the student requires to actively participate in interactions of increasing complexity with other individuals, objects and symbols in the learning environment. This interaction with the environment occurs on a regular basis and over time. The development is dependent on the environment; however, the student is not a passive recipient but takes an active role in their own experiences and development. The analysis was conducted by the independent first author, but to ensure inter-subjectivity, the senior author (S.B.) reviewed the consistency of the identified themes and structures of the themes for consensus. Data were first analyzed for each group of participants (students, teachers and school leaders) separately, followed by analyzing SSGT and social activity control group by participant groups (students and teachers) separately, after which segments from the transcripts were coded and tentatively and deductively themed by experienced facilitators. Moreover, the barriers of training, perceived social behavior change and aspects of training implementation were all indicative of social validity constructs. Data analyses are first presented for the overall pooled, multi-perspective thematic structure for participants reporting on SSGT, and then separately for the different groups of informants (students, teachers and school leaders), aligned with the overall themes. The thematic structuring followed the recommendations by Wolf [76]. The concept of social validity was operationalized by indicators of satisfaction, acceptability and feasibility. According to Wolf, social validity must be validated at several levels: the social significance of the goals, the appropriateness of the procedures and the social importance of the effects, all necessary for user satisfaction. Responses from the social activity control group are added after the responses from the training group. Thereafter, we present the comparisons between real-life versus digital experiences among all participants.

## 3. Results

### 3.1. Social Validity of SSGT—Thematic Structure

Figure 2 shows the thematic landscape for the pooled, multi-perspective findings. Social validity of study participation in terms of satisfaction, acceptability and feasibility were indicated by four overall themes: facilitators, social behavior change, barriers and implementation. We identified 19 subthemes for the themes: six for facilitators, four for barriers, five for social behavior change and four for implementation (see Figure 2 for the complete thematic structure). Under each theme of social validity, we first present the results from the SSGT group, followed by the results for the social activity control group. Generally, there were far fewer verbal data available for the social activity control groups than for SSGT for different questions and respondents after the interviews, as SSGT was perceived as more novel and complex and provoking more thoughts and experiences. Not all respondents in the activity groups provided experiences to all parts of the interviews.

### 3.2. Facilitators

#### 3.2.1. SKOLKONTAKT

Adolescent students with NDDs described their participation in SSGT as enriching, as it meant acquiring new social tools and interacting with other youths struggling with similar difficulties. The school environment was described by all participants as a safe place for SSGT and they stated that relevant themes were addressed in the training. Adolescent participants felt that the groups increased their social awareness. Regarding the SSGT formats, group discussions and the chosen topics as well as using individual social communication goals were highlighted as especially valuable in this regard. Typical citations indicative of the facilitators were as follows:

“*The best part was probably the discussions. I have difficulties with talking to peers and express my opinion. I learned how to talk about different things and how to express my own attitudes. It is good to have this on a regular basis, to ventilate and try to be open.*”(Adolescent 1)

“*The discussions were good; they were about things youths worry and think a lot about. When you talk about it in a group and discuss thoughts and actions, then you don’t feel that lonely, and life doesn’t feel that hard. You are not alone with your thoughts. The themes were good.*”(Adolescent 2)

“*I felt safe in the group. The group size was good, and I liked the regular training.*”

“*My goals were to answer questions and to be able to talk in class and during oral presentations. I have practiced on things I feel uncomfortable with, so it has been of value for me.*”(Adolescent 9)

“*Ahh, it will be valuable to me in the future, if I go to university. It is good to have practiced raising your voice.*”(Adolescent 10)

Teachers in SSGT perceived the group setting with supportive adolescent peers as a strength that also seemed to positively impact the generalization of learned skills. Teachers experienced that the whole school’s social climate had improved during and after the training, and that they had developed a deeper understanding of the unique characteristics of each participating adolescent. They saw a more rapid social development in the school and more interactions among the adolescents. They also felt themselves more engaged and that they had been armed with new and specific tools to work with their students and gained a better understand of their student’s functioning:

“*You notice which students you can stretch; you learn that they’re actually not that fragile and that they like when you challenge them.*”(Teacher 1)

“*We got motivated when we saw the development in the adolescents. When we saw things happen in school it helped us to see the benefits of programs like this. Even if it takes time and is a little bit inflexible, it is worth it when you watch the adolescents’ interactions.*”(Teacher 2)

“*Ahh, we have a lot of knowledge already, but after this we have more insight in processes and things that are difficult for the adolescents, as well as how they think, and I think we use this new awareness unconsciously.*” (Teacher 1)

“*Some of the adolescents have been bullied before, and I hope they can remember this training and the good feelings and enhanced self-confidence, and that it will make them try out the skills in new social settings.*” (Teacher 4)

“*You improve the general social climate in school when the adolescents with most social impairments gain new skills, if you do that, you see changes in the individual as well as in the big group.*” (Teacher 2)

The school leaders viewed that school was an adequate, safe environment for SSGT and naturally enhanced participation for students even outside of the actual training. From a school leader’s perspective, teachers and students seemed to have gained novel skills and more knowledge at hand promoting self-efficacy. In addition, they reflected that stronger relationships/cohesion had developed between students and teachers over the course of SSGT. It appeared to the school leaders that the knowledge had manifested, would remain and hopefully be spread within their school. This could have a significant impact, such as the prevention of school absenteeism:

“*It is valuable because we have learnt new ways of working with improving the social environment and we have never before had such a friendly open group of social science students.*” (School leader 2)

“*Most of our students have social difficulties and the school can become very quiet and without interactions, so we want to continue with the training and want other schools to do the same.*” (School leader 1)

“*It feels like a relief in some way, it feels like several knots have been dissolved, and that we had students like isolated islands and even with destructive relationships. Now it is friendlier, they know how to talk to each other.*” (School leader 1)

“*The knowledge is spread within the school since we learn from each other, even if not all teachers were part of the training. We gain the same understanding.*” (School leader 1)

#### 3.2.2. Social Activity Control

Adolescents who participated in the social control activity groups appreciated being in the given group context and taking part in the social activities offered. They reported to have enjoyed getting to know new peers. The activities, both those with and without physical elements, were perceived as engaging to different degrees. There had been some controversies in the groups about how much time to spend on which kind of activities. Students did not report explicit or specific social behavior change associated with the activity groups:

“*If I have learnt something valuable? I don’t really know. Maybe that I now can make cupcakes counts, now I can bake that, but I don’t really know.*” (Adolescent 5)

Teachers perceived the elements of the control activity groups as likeable and appropriate for the age group. They highlighted the importance and effort needed to help the adolescents to get in touch with each other. Activities with a high load of social interactions were described as most valuable. The group activities were described as safe, giving students more social security, which in turn might impact on their well-being in school. The activities were perceived as requiring quite some extra time, preparation and co-operation among school staff.

“*Best parts? Ahh, we liked the physical activities, but outdoor activities were sometimes horrible with rain and bad weather, we tried, we know physical activities are good and especially important for this targeting group, they are inactive.*” (Teacher 3)

“*We are more comfortable now in providing social activities for the students, but with the education we will probably add new tools to our knowledge.*” (Teacher 5)

### 3.3. Barriers

#### SKOLKONTAKT

Students expressed a need for more training sessions and warm-up activities, and sometimes experienced difficulties in understanding or finding time to perform the tasks and formats of SSGT:

“*All things were challenging and hard for me, yes, but I try to have a positive view and it’s good to practice the things you can’t do, you will be a better person, I had to fight.*” (Adolescent 10)

“*Sometimes I did not have that much time to work towards my goals.*” (Adolescent 10)

“*Ahh, when we all have the same social difficulties, interactions can be challenging and hard, when we need to talk to each other.*” (Adolescent 12)

Teachers reported that they were challenged by the tight schedule and standardization of the training scheme, especially in the start, as in their experience the content was broad and inflexible, requiring careful preparations. Regarding the material, teachers found that the assignments for students were not always clear enough or easy to relate to. On the other hand, in the longer run, these preparations had expanded their knowledge and were described as rather valuable. Although the time-consuming aspect of SSGT was described as a potential barrier, there was also insight of the need of length and intensity for participants to achieve real change:

“*I mean it is no quick fix. We have conducted 36 sessions. Change takes time.*” (Teacher 1)

“*In the beginning I was shocked by all the theories, theory of mind and all the text, I thought the adolescents would quit. More flexibility would be good, now we followed the manual strictly.*” (Teacher 2)

“*The weekly assignments were a little bit difficult sometimes. It was confusing in the beginning how to do them, but the feedback from the researchers was good.*” (Teacher 4)

“*The concrete weekly assignments were the best, when the adolescents did things like going to the bank or ordering something at a café. We could not change the activities since we followed the manual.*” (Teacher 3)

“*Three times a week is a lot, it is time and cost demanding. And, ahh all the paperwork, it takes time to read and learn the content of the program, but with more flexibility in the program, and if school psychologist can be involved, it is possible.*” (Teacher 4)

“*The program could be shorter, but that is difficult too, it’s part of the thing, that it takes time. With less time it would have been stressful and not that good for the participating students.*” (Teacher 4)

School leaders reported that a barrier to SSGT feasibility was the extra time needed for its implementation in terms of the preparation of sessions and staff to the whole approach, especially in the start-up period. The program standardization with limited degrees of flexibility was experienced as unfamiliar to the school’s work style:

“*It takes time and has to be included into the schedule, but it is valuable and we believe we can use it in the future in more flexible forms. The teachers need deep knowledge. The manual might be good for more inexperienced teachers to use at the start.*” (School leader 1)

“*When organizing, there are several things to consider, it is expensive in the form of resources and our teachers that facilitate SSGT have fewer teaching hours. In the beginning, the teachers had a lot of content to learn, they were not familiar with the manuals, the feedback and possibilities to ask the researchers was valuable.*” (School leader 1)

“*We had a worried parent initially, she said her child couldn’t do some things, but the safe environment at school and the presence of other adolescents with the same difficulties helped and the parent was relieved.*” (School leader 2)

### 3.4. Social Behavior Change

#### 3.4.1. SKOLKONTAKT

Students experienced new patterns of thoughts and actions, new habits, new friends, enhanced awareness of social codes and better access to tools that helped handling social situations and interactions associated with SSGT. They perceived a coherent social context and a positive social environment at school, which contributed to increased school attendance. Enhanced self-awareness and less loneliness were particularly highlighted as positive social changes:

“*The training has contributed to, ahh, it has helped me to follow a rhythm and keep me on the right path when it comes to thinking processes and behavior patterns. It has helped me to create new habits and it is very nice to do it in a group, not alone. I like that the training was at school.*” (Adolescent 3)

“*Now I have contact with some peers, we can talk about what´s important, and I can get support from them without worrying about what other think. I feel very lonely and to experience that others have the same issues helps.*” (Adolescent 2)

“*I actually had some friends coming over to my birthday party for the first time. I used to watch people with friends thinking it would never happen to me.*” (Adolescent 6)

“*If this was valuable for me? I would say yes, definitely, ahh, we have more contact. That I now have contact with others and can talk about what’s important.*” (Adolescent 1)

“*For me, participation was a game changer—it saved me.*” (Adolescent 6)

Teachers perceived that students had developed skills to share opinions, talk in front of a group and show more self-assertiveness, which had also affected schoolwork and conversations beyond the SSGT situation, e.g., unstructured times of the school day and during leisure time. School staff described more happy, motivated and relaxed students with enhanced school attendance:

“*The students are more relaxed and you see them talk to each other and laugh. Some of them were wallflowers, if you don’t give them opportunities to practice they will remain silent.*” (Teacher 2)

“*Positive effects? I saw this great development in a student that I have followed, it is a huge difference. Before, she did not want to sit next to peers or talk to anybody. She has other body language now and takes initiatives. Now I see that she has friends and is happier and more relaxed.*” (Teacher 4)

“*The adolescents now have networks of friends from the groups. We see big changes among some of the adolescents. The group has helped them to build trust and they talk much more with each other.*” (Teacher 3)

“*The social goals have helped them and I think it affects the academic achievement too. You have adolescents with no motor and the new feeling with sense of context and enhanced participation, ahh and safety affect results in school.*” (Teacher 4)

School leaders expressed contentment with that they had perceived students to have improved on conflict solving skills, showed and appreciated social interaction with each other, viewed the emergence of friendships, enhanced self-awareness among adolescents and improved well-being. Generally, they perceived a better mood at the school among participants:

“*The students are more open and take initiatives to social activities. They have by themselves suggested social groups like gardening group or movie clubs. We did not see this earlier. They interact across age groups as well. It is pretty fantastic and would not happen last year. It spreads like rings on water, it benefits everyone.*” (School leader 1)

“*We cannot believe in some of the adolescents’ development and change. It is a big difference and this participant has different body language and seems so much happier.*”

“*These students are sometimes often absent from school and it is important to work preventatively and that they gain self-awareness and can practice what challenges them.*” (School leader 2)

#### 3.4.2. Social Activity Control

Adolescents in the control group enjoyed the social activities and having possibilities to meet other youths; however, there were no descriptions of gained skills or further developed friendships:

“*Concrete change? No, I don’t think so. I think of the Yoga session, and that my body is really stiff.*” (Adolescent 5)

“*If it has helped me? Well, perhaps the games helped me socially. When I see other people, I am more social compared to when I play games on the Internet. I suppose it is good to practice on it.*” (Adolescent 7)

### 3.5. Implementation

#### Skolkontakt

Adolescents appreciated that SSGT in school was practical, and that they could get help from group members and practice the new skills easily even outside of the actual sessions. One student was aware of the long waiting lists and bureaucracy involved with referrals to clinical settings and pointed out the advantage of all of this being unnecessary with SSGT in a regular school setting:

“*The advantages with social skills training in school compared to outside school is that it is very accessible, you don’t have to hassle with remiss to a place with long waiting list and such, and if you have issues with getting friends in your school, you get the grounds and have actually talked to people at school in your group and you can talk to them again.*” (Adolescent 4)

“*I have conducted social skills training before, but it is very easy in school, you don’t have to take all the responsibility since the training is in school with a set time during the week.*” (Adolescent 2)

“*The training in school is good, you feel safe, in child psychiatry you don’t know anyone, in school you might know someone in the group, ahh and it’s more familiar.*” (Adolescent 10)

Teachers believed both in the benefits of SSGT and that it was part of the school’s mission to engage in the social communication development of their students in ways such as SSGT, especially at schools which aim to achieve educational inclusion of many with NDDs. Teachers believed that there were many students in need of social skills training in a school setting:

“*We need to see these adolescents and exactly because many of them have difficulties with interaction and communication, we need this. It is important for the future, the social part is very, very important, and we want them to succeed after school.*” (Teacher 5)

“*This is realistic in the school environment, and it is a question of democracy and policy, if you could measure the long-term effects of the training you see the benefits for society. It is what results we measure, whether the academic achievement and marks are ok is one side, but then, what next, do we want the adolescents to just stay home and stare at the walls, what’s next for them?*” (Teacher 1)

“*We have integrated the training in the ordinary curriculum, it is tough and an expensive intervention, it could work with professionals from outside, however it is best with staff from the school, we know the students.*” (Teacher 2)

“*Our students might not have difficulties with school and achievement, the social environment is their big issue. You can use the networks from the groups and interactions are more smooth and natural.*” (Teacher 1)

School leaders reported that they had been looking for social communication programs suitable for their school and students for some time, that SSGT had been meeting their expectations and that they desired to continue with it even after the research project. They saw interventions such as these as part of their school’s responsibility for democratic development and social inclusion. They had tried several informal unstandardized approaches without being convinced but desired to implement an evidence-based program, such as SKOLKONTAKT^®^, with sound theoretical principles. School leaders stressed the importance of financial support during the implementation phase, transparency and continuous exchange with the clinicians/researches during that time. They appreciated that the clinicians understood the challenges and demands of the school environment. Moreover, the feedback and supervision from clinicians/researchers during the intervention was seen as valuable and necessary for implementation. Generally, SSGT was valued as a means of getting students involved in social communication and interaction conventions and expectancies, which was hard to achieve in regular education otherwise, despite being a paramount prerequisite for a good school life for everyone:

“*Students with NDDs as well as other students with social impairments or anxiety need to learn strategies, and they are at all schools. We have seen a need of practicing social skills. We have often solved conflicts afterwards. It is not always in the schedule and curriculum and we see this as a complement. It has been a driving force.*” (School leader 1)

“*We had been looking for how to develop and strengthen the school and looked for something that was evidence-based and could work. The activities conducted in the control group are social activities we have done before.*” (School leader 1)

“*It is important that the researchers understand the school environment, everything has to be transparence from the beginning. It is a big project and researchers have to understand what is reasonable to request.*” (School leader 2)

### 3.6. Real-Life Training vs. Digital SSGT or Social Control Activities

Students reported to be satisfied overall with the transformation of both face-to-face SSGT and the social control activities into digital formats, and appreciated that SSGT and control activities continued, despite the ongoing pandemic. It was perceived that the majority of the SSGT contents could be addressed on-line. However, adolescents themselves expressed the real-life sessions to be more valuable and experienced that they were missing out on the social encounters with their peers, particularly as it hindered true social interactions and the development of friendships:

“*For me it worked digitally, but it is better to meet for real. It was a little bit like you did not see each other for real. It would have been better to meet physically, but there is not much you can do about it.*” (Adolescent 9)

“*It was quite ok, clearly it would have been more fun to meet in person. I have not immediately made any friends after joining, it is harder to get in touch maybe when it’s online. Some things can be done but others not. The physical activities were hard to conduct.*” (Adolescent 11)

“*Oh, first of all, it was boring not to meet face to face, you missed the social part, but otherwise I think it has worked surprisingly well. It was not that fun, but you have to make the best out of it.*” (Adolescent 10)

Teachers still observed individual improvements of social communication skills following digital SSGT, but much limited generalization to outside of SSGT situations and social behavior change. Teachers also highlighted the diversity among adolescents in how they could take advantage of the digital sessions in both SSGT and control treatment. It seemed that they suited some very well and others not at all, and that this did not correspond to the analogue sessions. Teachers conducting the social control activities found it challenging to translate all social activities into adequate digital replacements, especially the physical ones:

“*When transformed to digital, the physical activities were more difficult, however we tried digital walks and communicating online with the adolescents. We could talk to them, but you are more sensitive when you meet them.*” (Teacher 4)

“*We saw some changes after the digital training, some things like relief and that the students could somehow breathe more freely and that it was easier to be in the same room.*” (Teacher 5)

“*The digital format works for some and doesn’t for others, for some it is shit. You see the development in some, while others make less progress.*”(Teacher 5)

“*For students with big social stress and anxiety, who stay home from school, it can be very urgent here, and it can be easier to start online.*”(Teacher 5)

“*Ahh, in other words it was much easier the last time when we had physical meetings, when you saw the group, you saw progress in just a couple of weeks, I believe it is important to be seen.*” (Teacher 4)

“*I see differences from now when it’s digital compared to IRL. We could see more concrete change after last session with physical meetings. We saw that the students actually talked to each other and you as a teacher could talk to them informal more freely at lunch and recess, which had not been possible before. Now when the training was digital, we see more individual progress than interactions. We see that the students have more knowledge and understanding, but I don’t believe in enhanced interaction effects when the training is online.*” (Teacher 2)

## 4. Discussion

Despite being a promising avenue to increase educational inclusion and build capacity in society, research on explicit and stand-alone SSGT in adolescents with NDDs and teachers providing the training in mainstream school settings has been scarce. The latter has been pointed out in a recent meta-analysis, as well as an international research and community strategy manifesto [30,60]. Little is still known about whether SSGT is perceived as more useful than other structured social activities in a school context, and if those who receive and deliver SSGT, as well school leaders in charge, view it as feasible, acceptable and providing satisfactory outcomes. Therefore, the objective of this study was to examine the social validity of SSGT compared to other social group activities in a mainstream context administered by teachers to adolescents with NDDs. Owing to the COVID-19 pandemic, we also examined how the target groups experienced real-life and digital SSGT formats of delivered training.

There was agreement among the different school stakeholders that SSGT for adolescents with NDDs is a largely socially valid. Both adolescent students receiving SSGT, school staff delivering it under the supervision of clinicians and school leaders being responsible for implementation perceived that the training was accessible, relevant and doable, well-placed in a natural daily environment and within the school’s responsibility. Thus, SSGT equipped the students and teachers with novel skills and knowledge. The acquired skills, such as making one’s voice heard, knowing and daring to initiate small-talk and managing challenging social situations are benefits from the training. On a generic level, these acquired skills improved social behaviors, school performance, school attendance, motivation and the overall social climate at the school. SSGT formats that were experienced as extra valuable were group discussions and identification and pursue of individuals goals. On the other hand, particularly initially, students and teacher also found parts of SSGT demanding in terms of complexity, inflexibility, intensity and added workload. School leaders stressed that extra economic resources and supervision from clinical professionals was key to implementation. Other comments regarding SSGT included a wish for more warm-up activities to precede the sessions, more clarity about some of the tasks and formats, and continued training beyond the standard number of sessions. Importantly, the social activities offered in the control group, while being appreciated and described as enjoyable, were not experienced by the participants as leading to noteworthy gain in skills, social behavior change or structural improvements at the school. The digital variant of SSGT that had to be introduced due to COVID-19 restrictions was perceived as a valuable alternative, but real-life SSGT was clearly preferred.

Our results are consistent with previous—mostly quantitative—research on social skills training in various educational setting and groups of students, including those with NDDs, indicating benefits of such actions in the areas of students’ social and emotional skills, attitudes towards school and academic performance [10,47,54,55,56,57,58,61]. Especially the stakeholders’ experiences were consistent with earlier studies pointing out the significance of working with peers in a natural context [80], and the natural context providing enrichened and immediate possibilities to practice, refine and generalize skills [58].

Still, the current study adds crucial value to the existing knowledge in several ways, as it indicates that explicit SSGT is a socially valid method in the eyes of key school stakeholders, when delivered by regular mainstream school staff supervised by experienced clinicians, with some available additional resources for implementation. Although this is a qualitative study with a focus and lived experience, it is also among the few studies allowing to contrast the specificities of SSGT versus other social group activities. Attitudes of stakeholders are likely to determine at least as much as hard scientific evidence, whether SSGT in mainstream school setting will be successful and sustainable. It is not a matter of course that staff is positive towards evidence-based practices. For instance, research from Swedish preschools yielded reservations by staff implementing scientifically supported practice [66]. Our study is also among the first to examine how SSGT is perceived by adolescents with NDDs, a group that is likely to benefit more from such forms of training than younger children [44]. It is particularly encouraging that teachers were positive about delivering SSGT, as it has been demonstrated that effects of social training may be greater in school when delivered by staff [10]. Moreover, there are potential spin-off effects from training students on the teachers’ own skills and knowledge about how to approach diversity, which enhance teacher self-efficacy [81] and decrease the risk of teacher burnout [82].

The COVID-19 outbreak presented not only a challenge, but also an opportunity to explore how face-to-face standard SSGT SKOLKONTAKT^®^ was experienced in comparison to a digitally delivered variant of the same training. While there is evidence that students can improve social behaviors in distanced education [83,84], to the authors best knowledge, this is the first study to examine the perception of digital SSGT in school settings. Participants appreciated that SSGT could continue despite pandemic-related restrictions, and the digital variant was described as useful and a viable alternative to face-to-face training. Still, there was robust consensus that the digital variant missed out on the essential parts of personal meeting and proximity, the experience of group coherence and spontaneous additional social opportunities before and after the sessions. Thus, digital variants of SSGT appear to be an important addition and alternative to standard real-life training to increase accessibility to services in an increasingly digital world, but apparently, they are not preferred over real-life SSGT. Controlled trials are needed to establish the effects of digital SSGT on relevant outcomes, such as acquired social skills, achievement of social goals and quality of life.

There were several barriers to SSGT and its implementation described by the different groups of participants. Adolescents demanded a slower start to the training and its sessions, as well as more clarity on expectancies and the contents of some tasks. SSGT SKOLKONTAKT^®^ already includes a high degree of clarity, repetition and information. There are also separate information meetings before the start of the training, such as warm-up opening rounds in each session. As these barriers were mostly limited to the initial phase of SSGT, such experiences might be hard to avoid completely when introducing a new program. Additionally, the teachers reported that they experienced high demands put on them in the beginning, but these diminished after some sessions, as they got more familiar with the training and their role. Teachers also experienced that SSGT was dense, quite labor intense and inflexible. Again, when introducing a novel and highly structured program to staff not familiar with conducting such training, it might be hard to avoid that school staff experience inflexibility of a program, additional burden, responsibility and complexity. School leaders stressed the need for extra economic and personnel resources to make possible the implementation and sustainability of SSGT. Both school leaders and teachers also pointed out the need for supervision from experienced clinical SSGT trainers. Recurrently, the literature shed light on the importance of interdisciplinary collaboration, e.g., more participation of school psychologists and other professionals with expertise in NDDs in educational programming [85,86,87]. Therefore, the successful implementation of SSGT in mainstream school settings is complex and requires careful planning, collaboration between educational and clinical services and additional resources by authorities. With a holistic view on inclusion, individual needs as well as surrounding components are equally important. This is described by Bronfenbrenner [79], where several levels of the social system affect the situation for a student in need. Within the bio-psycho-social approach, all levels of the social system interact and influence the individual. In this study of social validity, we not only addressed the individual and personal development, but also its immediate and remote social environment.

Participating adolescents mentioned that it was primarily the group discussions and pursuing of personal social goals had helped them in social awareness and skill development. Previous clinical studies on SSGT KONTAKT^®^ showed that particularly homework assignments and parent involvement were perceived as facilitators of training effects and the likelihood of generalization [50]. One can speculate that these differences are attributable to setting, with the school setting naturally offering more immediate options to practice and make available peer support and pressure, compensating for parent support. Several of the adolescent participants inquired about continued SSGT beyond the 12 weeks program. Clinical SSGT KONTAKT^®^ has demonstrated markedly lager effects for 24 weeks as compared to the 12-week version of the training [75], which is why the development and evaluation of a longer version of SKOLKONTAKT^®^ might be reasonable. However, due to fatigue and ceiling effects, longer SSGT does not necessarily entail better results. Moreover, there might be limits to the school’s capacity, and there is always a certain risk in SSGT that participants’ social development concentrates to strongly around the SSGTs, although this risk seems lower when conducted in a natural environment. This is consistent with some adolescent students in our study expressing a wish for involving new and additional individuals from outside of the actual training group and saw the need of extending the formats of the SSGT to more challenging social settings. Unfortunately, SSGT involving tasks targeting the broader social environment of participants remain scarce [88]. However, social skills training in school for adolescents could be followed by transition (to adulthood) aids addressing diverse areas of life in order to offer broader help and achieve continuity of support and preparation for challenging areas and periods of life beyond school. Such interventions can be offered in various settings, have been found to be feasible, effective and desired by clients with NDDs and are in line with services that caregivers wish to be offered for their children [89,90]. Social psychology research has shown that participants often demonstrate improvements in their social behaviors following skills training, but these are not always aligned with raised sociometric status [91]. Therefore, to reach better social equity for students with NDDs in the longer term, additional measures are needed on top of social skills training alone.

This is a reasonably sized qualitative study showing a credible and meaningful pattern of results. However, we only examined a single school with motivated staff and previous experience of the target group. Thus, findings need to be treated with necessary caution and generalizability is limited. While we are confident that social validity of SKOLKONTAKT^®^ applies to other educational contexts, we strongly recommend independent confirmation of the social validity of the program in other school environments.

## 5. Conclusions

To make educational inclusion for students with NDDs a reality is a goal internationally and a priority in the NDD community, but even in high-income countries, this mission is still far from being accomplished [33,34,87]. SSGT delivered in mainstream schools by regular school staff may be an option to empower students with NDDs to gain control over their life at school and navigate social interactions with their peers and school staff more safely. It may also be connected to improved teacher skills in understanding and approaching their students and foster the development of better relations between students and staff. This study indicates that both students, teachers and school leaders experience the effects of SSGT at school largely this way, and the method as socially valid across its characteristics and prerequisites. Although barriers and challenges to implementations are encountered, these are manageable and do not outweigh SSGT benefits. The same appears to be true for digital SSGT, although to a lesser degree, as face-to-face SSGT is clearly preferred by stakeholders. Other untargeted social group activities do not appear to compare to the experiences made by stakeholders involved in SSGT. We conclude that a broader implementation of SSGT in regular school settings for students with NDDs appears meaningful.

## Figures and Tables

**Figure 1 ijerph-19-01487-f001:**
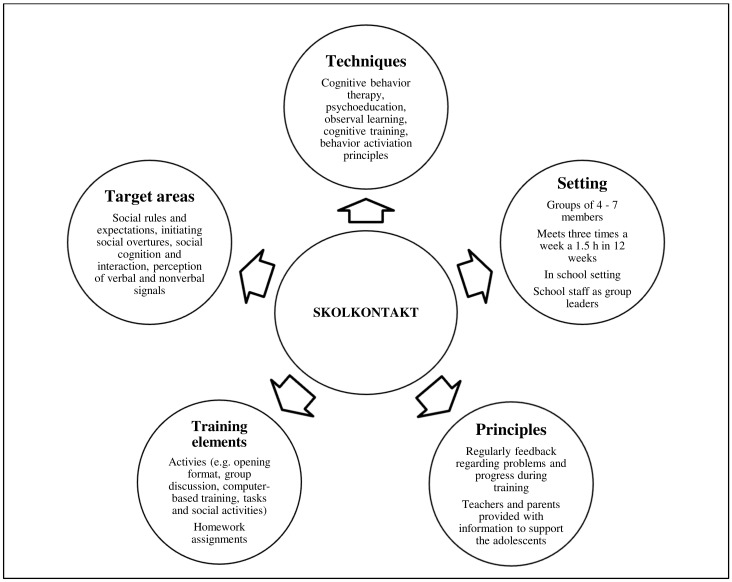
Characteristics of SKOLKONTAKT^®^.

**Figure 2 ijerph-19-01487-f002:**
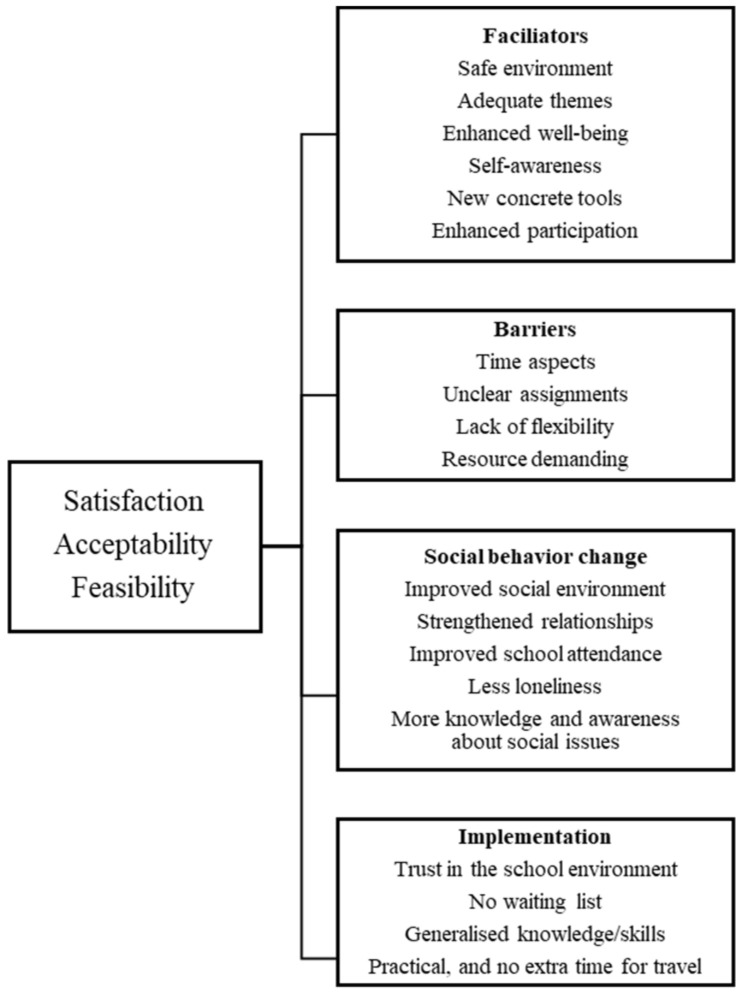
Thematic structure of social validity in social skills group training.

**Table 1 ijerph-19-01487-t001:** Student participant characteristics.

Gender	Age	Primary Clinical Diagnosis	Co-Existing Diagnosis/Symptoms	Intervention Group	Interview Setting
Female	20	ADHD	Social anxiety	SKOLKONTAKT	School
Female	19	ASD	-	SKOLKONTAKT	School
Diverse	18	ADHD	ASD, dyslexia	SKOLKONTAKT	School
Male	18	ADHD	ASD	SKOLKONTAKT	School
Male	20	ADHD	ASD	SKOLKONTAKT	Digitally from school
Female	18	ASD	ADD, OCD, Social anxiety	SKOLKONTAKT	Phonecall
Female	18	ADHD	ASD	Activity group	School
Male	17	ASD		Activity group	School
Diverse	17	ASD		Activity group	School
Male	17	ADHD	ASD	Activity group	Digitally from school
Diverse	17	ASD	ADHD	Activity group	Digitally from school
Male	17	ASD	Anxiety	Activity group	Digitally from school
Female	19	Social anxiety	Subclinical autistic symptoms	Activity group	Digitally from home

Note. ASD = Autism Spectrum Disorder, ADHD = Attention-Deficit Hyperactivity Disorder, OCD = Obsessive Compulsive Disorder.

**Table 2 ijerph-19-01487-t002:** Training formats in SKOLKONTAKT^®^.

Training Format	Description	Objective
Mandatory		
Agenda	Detailed and visualized information of the session, the structure and activities.	Give a clear structure and create safety through routines and transparency.
Opening session	Round of introductions. Expressions of feelings (visual support in emotion-figures as scaffolding structures). How do people feel today? Do they have something to share? Are there wishes for today’s session?	Warm-up activity, initiate contact, build safety, start interactions and give the opportunity to express mood, feelings and motivation.
Closing session	Evaluation of the session and recap of the day. Sharing positive and negative experiences with the activities and tasks. Round of evaluations.	Promotion of interaction between group members. Practicing talking in a group and remembering names and endorsing social overtures.
**Recurring**		
Snack-time	Interaction in a non-structured situation.	Practice small-talk, practice turn-taking and encourage social skills use.
Group rules	Rules are formulated and founded by the group. Rules are visualized and made concrete for adolescents. Examples of rules: listen actively to each other, secrecy, only give positive feedback and use kind language.	Building safety and trust in the group and treating each other with respect. Creating an environment where adolescents dare to talk, open up and share experiences.
Homework assignments	Setting and perusing meaningful individual and general social communication goals. Examples of individual goals: ask questions, handle stressful situations, understand nonverbal signs and make an appointment with a classmate.	Building and training skills outside of the training context, behavior activation and generalization of skills to everyday situations. Personalization of social skills training.
Coaching sessions towards assignments and individual goals	Assignments are followed up and feedback is given by experienced facilitators. Barriers to goal attainment are analyzed and elements for goal completion are established. Group trainers give constructive feedback towards the assignments.	Reinforcement of social successful behaviors or receive suggestions on alternative approaches to goal attainment. Fine-tuning of personal goals. Opportunity to talk about own social experiences regarding concrete actions.
**Variable**		
Group activities	Baking together, cooking, playing sports and visiting a café or a museum.	Group cohesion and practicing cooperation and social skills in an informal setting.
Group discussions	Discussion of specific topics, e.g., to have social contact, recognize social situations in school, have feelings of loneliness, issues and typical problems of being young and to handle changes, misunderstandings and conflicts in school and life.	Exchanging experiences, social cognition, social relationships, feeling safe in a social setting, daring to raise your voice and have an opinion, sharing advice, practicing active listening and learning ways to handle challenges, stress and emotions.
Group exercises	Group games, social interaction games, how to handle stressful activities, watching the Movie for the Assessment of Social Cognition and facial affect recognition training.	Developing strategies for difficult social situations. Practice and discuss real-life situations, improve social thinking and detect socially relevant nonverbal signs.
Role-play	Participants play and practice challenges and real-life scenarios, where the group members discuss the situations and actions of the protagonists to find solutions and an action plan.	Mimicking and solving different social scenarios in a safe environment so that they are able to handle social situations better in real life and school.

**Table 3 ijerph-19-01487-t003:** Interview guides by the respondent group.

Students	Teachers	School Leaders
Which elements and contents from the training */activities ** do you recall?	Generally, what do you recall from the training/activities?	How did you receive information about the training/activities and the research project?
Which parts of the training/activities did you like the most, and why?	Which parts of the training/activities did you like the most/the least, and why?	Do you think there is a need of the training/activities in school settings?
Which part of the training/activities did you like the least, and why?	Are training/activities like this appropriate as part of your work at school?	Is the training/are the activities appropriate for your school setting?
Is there anything in the training/activities that you would have liked to train/do more or less? ***	Is there anything about the training/activities that you would have liked to focus on/do more or less? ***	In your role as school leader, what did you need to consider and had to arrange to implement the training/activities at your school?
What did you think of the group discussions? ***	What did you think of the group discussions? Were the themes in the discussion appropriate and valuable? ***	In which way have the training/activities positively and negatively influenced daily life at your school?
What did you think of the training homework? ***	What did you think of the training homework? *	Can you see any changes among the adolescents or the teachers associated with training/activities?
Do you think some parts of the training/activities might have helped you? Which activities and why?	Do you think some parts of the training/activities might have helped the adolescents? Which activities and why?	Is it realistic and possible to implement training/activities at your school in the future?
Do you think the training/activities have improved your social skills? In what way?	Have you seen any enhanced interactions or improved social behaviors in the adolescents following training/activities?	What is important to consider for implementation of training/activities? What training, resources and support do your staff need for implementation?
Are there any concrete or specific changes, positive or negative, in your life that you think are due to the training/activities?	Are there any concrete or specific changes, positive or negative, that you have observed or noticed, that you think are due to the training/activities?	Are there any areas of possible improvements from your view according to the whole process and co-operation with researchers?
Do you think participating in the training/the activities will give you long-lasting improved social skills in life or in school? If so, in what way?	What do you think of long-lasting effects after the training/activities? Are there any? Have you seen any?	Which parts of the training/activities do you think are valuable or less valuable for your school?
Is there anything that could be better or made differently in the training/activities?	Is it possible and realistic to conduct training/activities like these in school in the future?	Do you think the training/activities have any spin-off effects for the adolescents in school and outside?
Were there enough, too many or too few training/activities sessions?	What do you think of the amount of the training/activities’ sessions?	Do you think the number of sessions of the training/activities were appropriate?
What do you think of the fact that this training is in your school? Is it positive or negative? Have you taken part of training/activities like this before somewhere else? Did the training/activities put an additional burden on you?	Do you think you have gained more knowledge and tools to help and understand your students, to develop the adolescents’ understanding of others, to develop the acceptance of the adolescents among others, to motivate and teach the adolescents to strengthen social interaction, to modify your teaching in order to help students to reach their goals and help the students to develop self-esteem?	Do you think your teachers have gained more knowledge and tools during training/activities to help the students to develop skills and reach social goals and other achievements?

Note. * The training = Social Skills Group Training (SSGT), ** The activities = social activities control intervention, *** only in students and teachers receiving or conducting SSGT.

## Data Availability

Raw qualitative data from this research are not publicly available, as this is part of the ethical approval. Data can be provided by the authors upon legitimate request.

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
