# Peer review of "Social Skills Group Training for Students with Neurodevelopmental Disabilities in Senior High School—A Qualitative Multi-Perspective Study of Social Validity"

_ijerph, 2022, doi:10.3390/ijerph19031487_

Round 1

Reviewer 1 Report

Thank you for providing an interesting paper. The inclusion of the voices of the participants in your social skills/social capacity building intervention makes a particularly important contribution to this field of study. 

In introducing the context for this research project you do raise some interesting points, but I believe you need to further elaborate on the importance of an inclusive social environment. Where you refer to the individual level characteristics that lead to negative outcomes, it sounds like the responsibility is on the individual to change to fit the expectations and norms of the school. I would argue this is not an inclusive school environment, but rather an environment of cultural homogenisation.

Your qualitative methods are well described, with only minor details needing to be added to give the reader a full understanding of the conditions in which your data was generated. I found the tables to be particularly helpful in communicating the assumptions and theoretical lenses through which you explored intervention. That said, I would add more detail in the body of your text about these theoretical assumptions, as they frame the way through which you understand intervention. For example, you refer to this program being informed by CBT but you don't elaborate on the importance of this.

Your discussion highlights the key contributions this study has made across a range of areas. Please see my annotated PDF for specific feedback but I would careful to not overstate the generality of your findings beyond this single context due to the small sample size and the use of participant perception as your instrument measuring outcomes.

Overall your paper is well constructed and mostly supported by appropriate references to current literature. Again, please see my notes for minor suggestions on how to improve the clarity of certain small sections. Thank you for an interesting report on your study and I look forward to reading your subsequent papers.

Reviewer 2 Report

I would like to congratulate with the authors for the quality of the work presented! Really good structure, clarity and puctulality of the methodology and procedure.

There are minor changes that would increase the overall quality of the manuscript:

  • you mention and refer to social validity theory in the introduction; a wider explanation of the theory, for instance with a figure,  would be useful to better understand the structure of your results when you show them.
  • You refer also to BrofenbrennerTheory at some point of the methodology for interpreting your results but there is no mention of the Theory in the introduction neither in the discussion, I would suggest to not reporting it if you don't actually use it in your work.

Congratulation! great job!

Reviewer 3 Report

Thank you for this important paper reporting on your research on social skills training in schools for young people with Neurodevelopmental Disabilities. Overall it is a very good paper that reports the research clearly.

There are some areas where editing for English language is needed but these are minor. 

In the results section some themes need to provide more evidence from the adolescent participants eg 3.2.2 and 3.3 have more quotes from teachers than adolescents. 

In the discussion it would be interesting to return to Brofenbrenner which offers an ecological framework for understanding the place of the social skills training within the lived experience opportunities of the adolescents. This section and the introduction would benefit from bringing in a more social-ecological understanding of neurodevelopmental disability to align better with the relational model of disability and recent literature on positioning neurodevelopmental disability within the bio-psycho-social model, given the focus of the paper on social skills development.
